

# Establishing reference values for age-related fecal calprotectin in healthy children aged 0–4 years: a systematic review and meta-analysis

Junxiang Zeng[1,*], Wenxian Yu[2,*], Xiupan Gao[1], Youyou Yu[1], Yunlan Zhou[1] and Xiujun Pan[1]

[1] Department of Clinical Laboratory, Xinhua Hospital, School of Medicine, Shanghai Jiao Tong University, Shanghai, China
[2] Department of Medical Genetics, Hunan Provincial Maternal and Child Healthcare Hospital, Changsha, China
* These authors contributed equally to this work.

Corresponding authors
Yunlan Zhou,
zhouyunlan@xinhuamed.com.cn
Xiujun Pan,
panxiujun@xinhuamed.com.cn

## ABSTRACT

**Background and Aims:** The use of fecal calprotectin (FC) as a biomarker in children under 4 years of age is limited, as widely accepted reference values have not yet been established. Thus, the present meta-analysis was performed to establish convincing age-related FC reference values.

**Method:** We conducted a comprehensive meta-analysis to establish age-specific reference intervals for FC in children aged 0–4 years by synthesizing data from multiple studies. An exhaustive search of three major databases was conducted from their inception through April 2024, and this was complemented by manual screening of reference lists to ensure the inclusion of all relevant studies. The eligibility criteria were restricted to English-language publications reporting quantitative FC levels in apparently healthy children under 4 years of age. A stringent study selection protocol was applied, with each article independently reviewed by at least two investigators to ensure accuracy in study inclusion, data extraction, and methodological quality assessment. For the statistical analysis, random-effects meta-analysis models were constructed using restricted maximum likelihood estimation to generate pooled mean FC levels and corresponding 95% confidence intervals (CIs) at the study level.

**Results:** A total of 23 studies ($n = 2,883$) were included in the qualitative synthesis. The pooled mean FC reference values (µg/g) by age group were as follows: <1 month: 257.70 (95% CI [189.70–325.70]), 1–6 months: 239.46 (95% CI [181.17–297.75]), 7–12 months: 115.72 (95% CI [89.69–141.75]), 13–24 months: 104.70 (95% CI [61.96–147.44]), 25–36 months: 75.18 (95% CI [43.94–106.42]), and 37–48 months: 33.89 (95% CI [24.43–43.35]). Assay methodology, particularly enzyme-linked immunosorbent assay, demonstrated significant heterogeneity ($p = 0.011$) in manufacturer-specific analyses. Furthermore, geographical variation had a significant effect on baseline FC levels in neonates. In contrast, the type of feeding and mode of delivery did not show significant effects ($p > 0.05$).

**Conclusion:** We established normal reference ranges for FC in healthy children aged 0–4 years. Assay methodology and geographic factors significantly influence baseline FC levels, underscoring the need for careful interpretation of FC levels in clinical settings.

## INTRODUCTION

Fecal calprotectin (FC) levels are higher in children than in adults, especially during the first year of life, before gradually decreasing in the subsequent years (*Roca et al., 2020*; *Kolho & Alfthan, 2020*; *Zhu et al., 2016*; *Oord & Hornung, 2014*; *Li et al., 2015*). The elevated FC levels observed in neonates are believed to result from two primary factors: (1) immunological stimulation by luminal antigens in conjunction with the immature physiology of the neonatal gastrointestinal tract, and (2) increased transmucosal leakage due to heightened intestinal permeability, which tends to normalize with age (*Herrera, Christensen & Helms, 2016*). The bactericidal, fungicidal, and immunomodulatory properties of calprotectin may also contribute to the innate immune defenses of healthy infants during the early postnatal period (*Kim et al., 2021*).

In both adults and children over the age of 4 years, a cutoff value of 50 μg/g is commonly used as the upper limit of normal FC levels. However, no universally accepted cutoff values currently exist for children under 4 years of age, posing challenges to its use in routine clinical practice (*Zhu et al., 2016*). Despite numerous attempts to establish age-specific reference ranges for FC in pediatric populations (*Zhu et al., 2016*; *Oord & Hornung, 2014*; *Li et al., 2015*), the literature remains fragmented and inconsistent. These discrepancies largely stem from methodological variations in FC testing, particularly the lack of standardization in assay protocols. Different assay techniques, and even manufacturer-specific variations in commercial kits, yield significantly different results (*Pelkmans, de Groot & Curvers, 2019*; *Mirsepasi-Lauridsen et al., 2016*). In addition to methodological differences, considerable heterogeneity exists in the characteristics of study populations. Variations in age group classifications, ethnic composition, and geographic location further contribute to the inconsistencies in reported FC reference values (*Jukic et al., 2021*; *Juricic et al., 2019*). Such variability highlights the inherent difficulty in generating universally applicable reference intervals and reinforces the need for comprehensive research to systematically address these confounding factors (*Juricic et al., 2019*; *Padoan et al., 2018*). The lack of standardized FC reference values for children under 4 years of age is particularly problematic in clinical settings, where it may lead to misinterpretation of results, diagnostic uncertainty, and the risk of overdiagnosis and subsequent over-treatment (*Kapel et al., 2010*; *Haisma et al., 2020*).

Defining updated and reliable reference ranges for FC in newborns and infants—including both preterm and full-term neonates—as well as in older children up to 4 years of age, is therefore essential. Accurate interpretation of FC levels is critical for distinguishing between physiological changes and pathological inflammation, much like the clinical use of bilirubin (*Kapel et al., 2010*). In light of these issues, the present study aimed to systematically synthesize published data to establish normative FC reference values for healthy children aged 0–4 years through a comprehensive systematic review and meta-analysis, while also identifying potential sources of variability across the existing literature.

## MATERIALS AND METHODS

### Eligibility criteria

The investigators structured this systematic review in accordance with the Preferred Reporting Items for Systematic Reviews and Meta-Analyses (PRISMA) guidelines (*Cumpston et al., 2019*). This systematic review protocol has been formally registered with the International Prospective Register of Systematic Reviews (PROSPERO) (*Sideri, Papageorgiou & Eliades, 2018*) and assigned the registration number CRD42023473799.

### Search strategy

After conducting a thorough search of PubMed, Medline, Embase databases, and the Cochrane Library electronic databases, we retrieved original studies that reported FC concentrations in healthy children. The search was conducted using the keywords ("Leukocyte L1 Antigen Complex" OR "Calprotectin" OR "S100A8/A9" OR "MRP8/14") and ("Infants" OR "infants, Newborn" OR "Neonates" OR "Meconium" OR "Healthy children"), with the final update on April 8, 2024. The databases were subsequently combined and searched using Covidence—a web-based software platform—following the strategy outlined in the Fig. S1.

### Study selection

To identify relevant studies, two members of our research team (ZJX and YWX) evaluated the eligibility of each record in a standardized and unbiased manner independently. The initial screening process involved a chronological review of titles and abstracts, followed by the selection of studies after a thorough assessment of their full text and references. Any disagreements regarding the eligibility of studies were resolved through discussion with the third reviewer (ZYL or PXJ). However, there were no disputes between the two reviewers for study selection.

### Inclusion/exclusion criteria

Following a thorough evaluation of full-text articles, inclusion for quantitative synthesis was contingent upon the following criteria: (1) availability of sufficient data to obtain FC values numerically and the number of participants; (2) targeted aged 0–4 years of children and relation to FC studies; (3) state and/or indicate in "patients characteristics" that the participants were full-term, healthy newborns or children, with no evidence of gastrointestinal diseases or autoimmune disease; (4) clear description of the method employed for FC measurement.

Studies with the following publication types and study characteristics were excluded: (a) duplicate publications; (b) abstracts, review articles or case reports; (c) inclusion of preterm or very low birth weight (VLBW) infants as participants; (d) studies of calprotectin in other body fluids (*e.g.*, serum, plasma, amniotic fluid, placenta); (e) studies wherein infant age was outside the defined study age categories.

## Data items

Data extracted from each study included the following: (a) study attributes (last name of the first author and year of publication), (b) participant demographics (sample size and age range), and (c) information on fecal FC testing methodologies used in the laboratories; (d) type of feeding and the mode of delivery; (f) FC values including the mean, median, standard deviation (SD), interquartile range (IQR), minimum/maximum values range, and 95% confidence intervals (CI).

We defined five age categories: <1, 1–6, 7–12, 13–24, 25–36, and 37–48 months. For studies providing age ranges, but not specific means, we calculated the mean age to represent each category. As an example, for a 2–4-month age range, the mean age was considered as 3 months and the subjects were assigned to the 1–6-month group.

Extra attention was focused on cases with some missing data in the literature but present in graphics. To convert graphical data into the corresponding numerical format, we utilized the web-based "Web Plot Digitizer" tool (*Drevon, Fursa & Malcolm, 2017*), which has been established as a reliable method for executing such conversions.

## Quantitative data synthesis and analysis

In the meta-analysis, the normal approximation method was employed for estimating both the mean and the reference interval (*Németh et al., 2017*; *Siegel, Murad & Chu, 2021*). Recognizing that a key assumption of frequentist meta-analysis, particularly when applying normal approximation methods, is the ideal of homogeneity of within-study variances (*Siegel, Murad & Chu, 2021*), we opted for a random-effects approach. Since direct estimation of the 95th percentile's variance is not feasible owing to its calculation method, we employed an alternative approach based on asymptotic theory (*Matti et al., 2022*; *Brown & Wolfe, 1983*). This approach required data from studies that provided sufficient information on sample size and either median or mean with SD. A comprehensive decision tree diagram (Fig. S2) was accordingly prepared to illustrate the calculations performed using the employed mathematical formulas.

The reference interval was calculated as the mean ±1.96 SD. The confidence intervals for the mean and the boundaries of the reference interval were calculated using the normal approximation (*Bland & Altman, 1999*) and visually represented as diamonds in the forest plot and shaded areas around the endpoints, respectively.

## Risk of bias and quality assessment

Due to the absence of established methodological standards or guidelines for FC measurements and as this review was primarily descriptive, no formal risk of bias assessment was conducted. Study quality was evaluated using the Joanna Briggs Institute's Meta-Analysis of Statistics, Assessment, and Review Instrument (JBI-MAStARI) (*Moher et al., 2009*), a comprehensive tool designed to assess the methodological rigor and reliability of quantitative research.

## Statistical analysis

In our meta-analysis, we pooled the FC reference ranges as mean values along with 95% CI. Further subgroup analyses were conducted to investigate the possible influence of the feeding type, delivery mode, assay methodology, and geographical region on FC. To ensure robust and reliable outcomes, we employed a random-effects model for data analysis, and the model was estimated using restricted maximum likelihood estimation (RMLE). The Higgins index ($I^2$) statistic was used as a measure of consistency among studies. Statistical significance was defined as $p < 0.05$, and all tests were two-tailed. The entire statistical analysis was conducted by using STATA version 16.0 (College Station, TX, USA). The Python pyecharts library was utilized to create figure.

# RESULTS

## Search results

The detailed process of study identification, screening, and selection is illustrated in Fig. 1. An initial search of electronic databases yielded 1,689 records. After removing duplicates, the titles and abstracts were screened for relevance, resulting in 96 articles eligible for full-text review. Of these, 73 studies were excluded based on the following criteria: 47 did not provide sufficient information to estimate CIs; 12 involved populations outside the target group (*e.g.*, preterm or very low birth weight infants); eight lacked data regarding the number of participants within each age subgroup; and six included infants whose ages did not align with the predefined age categories (<1, 1–6, 7–12, 13–24, 25–36, and 37–48 months). After applying all eligibility criteria, 23 studies were included in the final quantitative synthesis.

## Study characteristics

The demographic details and FC-related characteristics of the included studies (*Zhu et al., 2016*; *Oord & Hornung, 2014*; *Li et al., 2015*; *Velasco Rodríguez-Belvís et al., 2020*; *Campeotto et al., 2004*; *Hestvik et al., 2011*; *Rugtveit & Fagerhol, 2002*; *Nissen et al., 2004*; *Song et al., 2017*; *MacQueen et al., 2018*; *Lee et al., 2017*; *Günaydın et al., 2020*; *Garg et al., 2017*; *Park et al., 2020*; *Rougé et al., 2010*; *Roca et al., 2017*; *Łoniewska et al., 2020*; *Li et al., 2014*; *Campeotto et al., 2021*; *van Zoonen et al., 2019*; *Laforgia et al., 2003*; *Jung & Park, 2020*; *Lisowska-Myjak, Skarżyńska & Żytyńska-Daniluk, 2018*) are summarized in Table 1. These 23 studies, published between 2002 and 2024, originated from 15 different countries and involved a total of 2,883 healthy children aged 0–4 years. Sample sizes varied considerably, ranging from 27 to 257 participants per study. All studies reported age-specific FC levels for at least one of the following predefined age categories: <1 month, 1–6 months, 7–12 months, 13–24 months, 25–36 months, and 37–48 months. The studies also varied in terms of participant ethnicity, assay methodology, and the distribution of subjects across age groups. In terms of data presentation, 13 out of 23 studies reported FC levels as medians accompanied by minimum and maximum ranges. Three studies

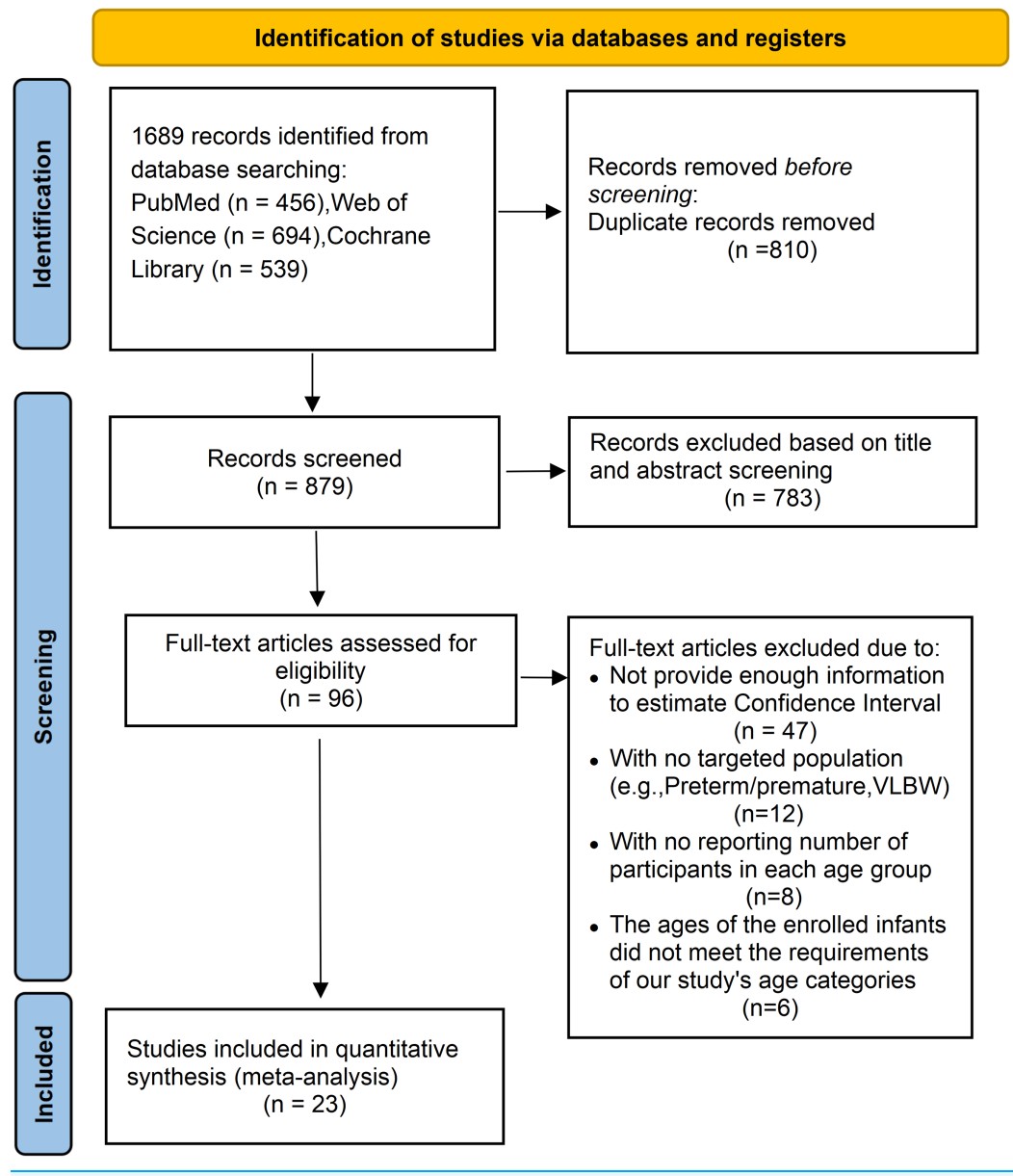

**Figure 1** **The PRISMA flow diagram displaying the selection of studies and reasons for exclusion.**

presented means with standard deviations (mean ± SD), four used medians with interquartile ranges (Q1 and Q3), and another three reported medians with 95% CI. In four studies, FC data were provided only in graphical form; these values were digitally extracted and converted into numerical data, which are presented in Fig. S3.

## Methodological quality

The quality of the included studies was assessed using the JBI Meta-Analysis of Statistics Assessment and Review Instrument. The results of this appraisal are provided in Table S1. Based on the JBI criteria, 10 of the 21 eligible studies met all 10 quality items and were

**Table 1 Characteristics of individual studies.**

| Author, Year | Location | Study design | Measurement method (Assay type, name, producer) | Descriptive statistics | Fecal calprotectin (μg/g), (Number of subjects) Age groups (months) | | | | | |
|---|---|---|---|---|---|---|---|---|---|---|
| | | | | | <1 | 1–6 | 7–12 | 13–24 | 25–36 | 37–48 |
| Roca et al. (2020) | Spain | Cross-sectional | LFIA, Quantum Blue fCAL, BÜHLMANN | Median (Q1, Q3) | 303 (210,412) (n = 43) | 325 (240,615) (n = 64) | 63 (45,171) (n = 46) | 97 (80,355) (n = 42) | 71 (50,180) (n = 45) | / |
| Campeotto et al. (2004) | France | Prospective | ELISA, Calprest, Eurospital | Median (Range) | / | | 167 (22–860) (n = 69) | | / | |
| Li et al. (2015) | China | Prospective | ELISA, fCAL ELISA, BÜHLMANN | Median (Range) | / | 282 (47,545) (n = 136) | 114 (6,937) (n = 102) | | / | |
| Hestvik et al. (2011) | Uganda | cross-sectional | ELISA, Phical, CALPRO | Median (95% CI) | 345 (195,621) (n = 14) | 278 (85,988) (n = 27) | 183 (109,418) (n = 27) | | / | |
| Rugtveit & Fagerhol (2002) | Norway | / | ELISA, Phical, CALPRO | Median (95% CI) | 269 (276,847) (n = 21) | 263 (185,618) (n = 20) | 78 (61,188) (n = 22) | 67 (63,191) (n = 20) | 64 (70,184) (n = 89) | / |
| Nissen et al. (2004) | Netherland | / | ELISA, Calprest, Eurospital | Median (Range) | 235 (172–2,880) (n = 27) | | | / | | / |
| Oord & Hornung (2014) | Denmark | Retrospective | ELISA, fCAL ELISA, BÜHLMANN | Median (Range) | / | 192 (29–537) (n = 9) | 72 (31–201) (n = 12) | 47 (29–223) (n = 20) | 31 (29–70) (n = 24) | 36 (29–70) (n = 8) |
| Zhu et al. (2016) | China | Prospective | ELISA, fCAL ELISA, BÜHLMANN | Median (Range) | | / | / | 18 (12–24) (n = 72) | 31 (24–35) (n = 50) | 40 (36–48) (n = 45) |
| Song et al. (2017) | South Korea | Prospective | FEIA, ImmunoCAP, Phadia | Mean ± SD | | / | | 145 ± 185 (n = 39) | 101 ± 176 (n = 30) | 22 ± 26 (n = 55) |
| MacQueen et al. (2018) | USA | Prospective | ELISA, Phical, CALPRO | Median (Q1, Q3) | 123 (51,193) (n = 38) | 148 (91,246) (n = 211) | / | | / | |
| Lee et al. (2017) | South Korea | Prospective | FEIA, ImmunoCAP, Phadia | Mean ± SD | 322 ± 318 (n = 43) | 197 ± 209 (n = 43) | / | | | |
| Günaydın et al. (2020) | Turkey | Cross-sectional | ELISA, IDK Calprotectin, Immundiagnostik AG | Median (95% CI) | 589 (123, 1,481) (n = 22) | 304 (20, 1,817) (n = 48) | 147 (10,802) (n = 14) | | | |
| Garg et al. (2017) | Australia | Prospective | ELISA, Phical, CALPRO | Median (Range) | | / | | 90 (17–445) (n = 12) | 67 (17–488) (n = 16) | 27 (17–46) (n = 24) |
| Park et al. (2020) | South Korea | Prospective | ELISA, Phical, CALPRO | Median (Q1, Q3) | 154 (70,346) (n = 146) | | | / | / | |
| Rougé et al. (2010) | France | Prospective | ELISA, Calprest, Eurospital | Median (Q1, Q3) | 138 (58,271) (n = 47) | / | / | / | / | / |

(Continued)

| Author, Year | Location | Study design | Measurement method (Assay type, name, producer) | Descriptive statistics | Fecal calprotectin (µg/g), (Number of subjects) | | | | | |
| --- | --- | --- | --- | --- | --- | --- | --- | --- | --- | --- |
| | | | | | Age groups (months) | | | | | |
| | | | | | <1 | 1–6 | 7–12 | 13–24 | 25–36 | 37–48 |
| Roca et al. (2017) | Spain | Prospective | FEIA, ImmunoCAP, Phadia | Median (Range) | 186 (20–2,964) (n = 51) | / | | 85 (8–843) (n = 17) | 45 (16–573) (n = 23) | 30 (20–851) (n = 20) |
| Loniewska et al. (2020) | Poland | Prospective | ELISA, IDK Calprotectin, Immundiagnostik AG | Median (Range) | 149 (12–681) (n = 74) | 109 (2–1,378) (n = 70) | 74 (3–1,447) (n = 64) | 59 (2–495) (n = 49) | / | |
| Li et al. (2014) | China | Prospective | ELISA, fCAL ELISA, BÜHLMANN | Median (Range) | / | 288 (35–937) (n = 54) | / | | | |
| Campeotto et al. (2021) | France | Prospective | ELISA, Calprest, Eurospital | Median (Range) | / | 88 (8–798) (n = 121) | / | | | |
| van Zoonen et al. (2019) | Netherland | Prospective | ELISA, fCAL ELISA, BÜHLMANN | Median (Range) | 332 (40–8,230) (n = 100) | / | | | | |
| Laforgia et al. (2003) | Italy | Prospective | ELISA, Calprest, Eurospital | Mean ± SD | 145 ± 78 (n = 131) | / | | | | |
| Jung & Park (2020) | South Korea | Prospective | FEIA, ImmunoCAP, Phadia | Median (Range) | 134 (11–2,000) (n = 228) | / | | | | |
| Lisowska-Myjak, Skarżyńska & Żytyńska-Daniluk (2018) | Poland | Prospective | ELISA, Phical, CALPRO | Median (Range) | 227 (34–1,067) (n = 20) | / | | | | |

**Note:**

LFIA, Lateral flow immunochromatographic assay; ELISA, enzyme-linked immunosorbent assay; FEIA, fluorescence enzyme immunoassay; Q1, 25th percentile; Q3, 75th percentile.
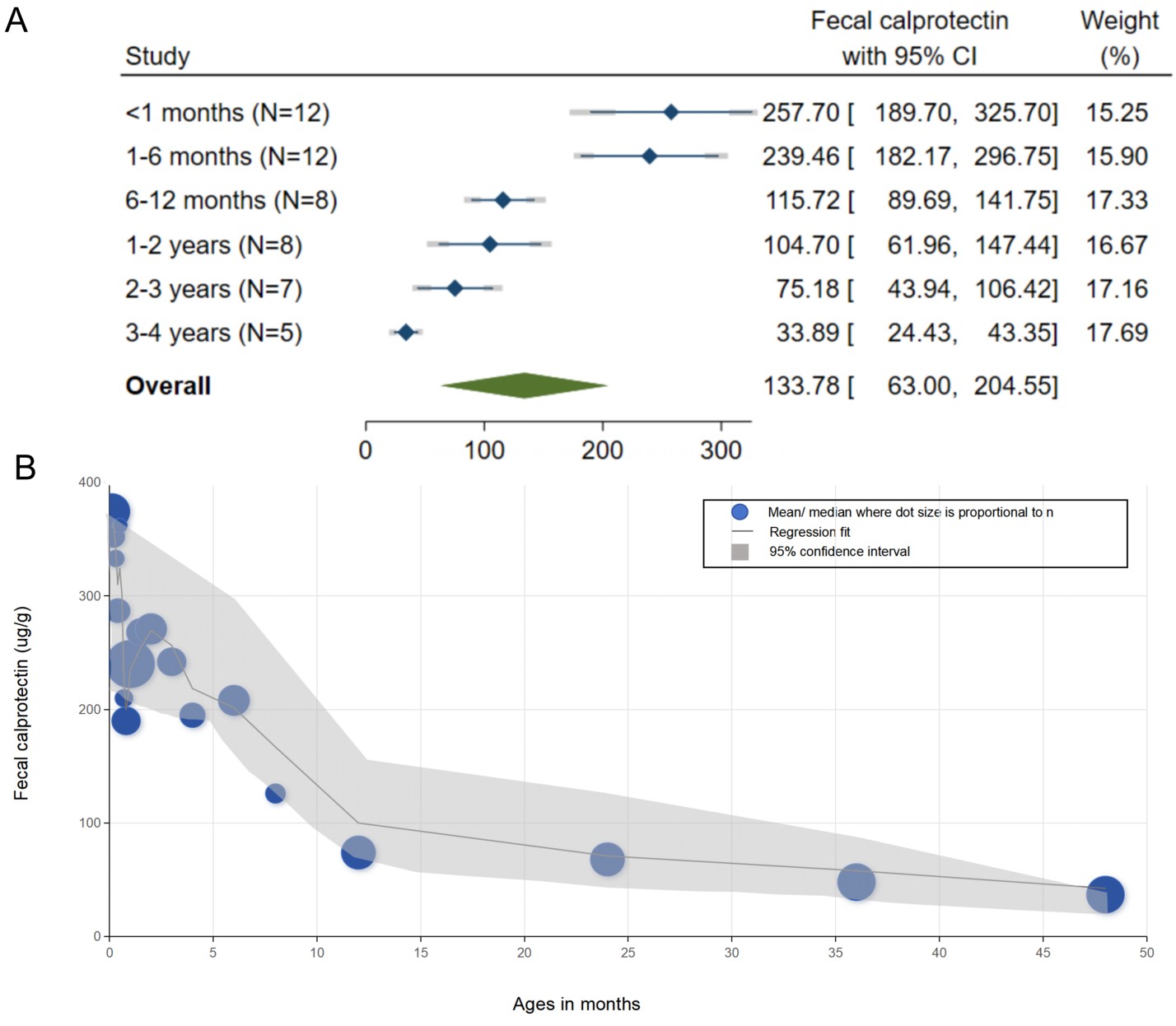

**Figure 2 Reference values of fecal calprotectin in children aged 0–4 years and their age-related trends.** (A) Forest plot of fecal calprotectin reference values by age groups; (B) Meta-regression plot analyzing age-related trends in fecal calprotectin reference values. It displays point estimates and 95% Confidence Intervals for the first 4 years of life. Dots represent mean or median values from the original reported studies.

classified as Level 1. The remaining 13 studies did not meet all criteria and were thus categorized as Level 2. This quality assessment highlights the varying methodological rigor across the included studies and was considered when interpreting the pooled results.

## Overall analysis

The reference ranges of FC for each age group are summarized in Fig. 2A. The pooled mean FC levels (μg/g) with corresponding 95% CI were as follows: <1 month, 257.70

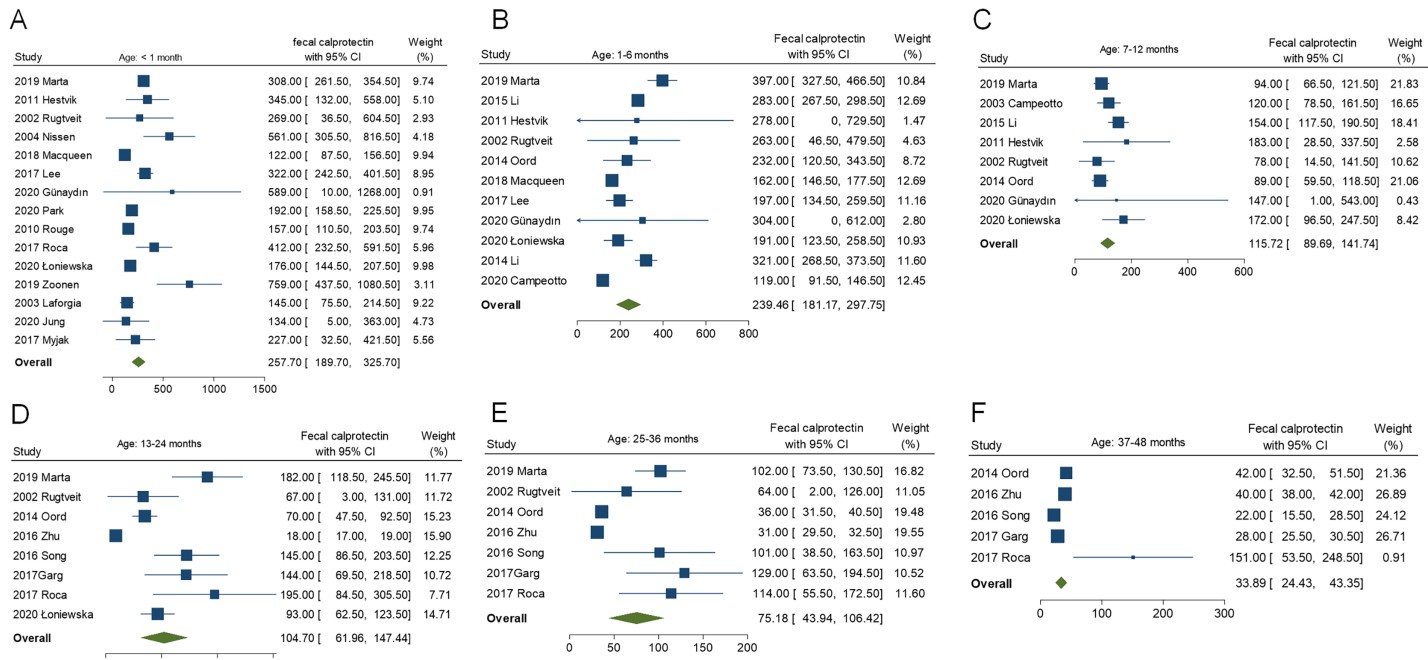

**Figure 3 Pooled fecal calprotectin reference values for each age subgroup.** (A) participants aged under 1 month; (B) participants aged 1–6 months; (C) participants aged 7–12 months; (D) participants aged 13–24 months; (E) participants aged 25–36 months, and (F) participants aged 37–48 months.

(95% CI [189.70–325.70]); 1–6 months, 239.46 (95% CI [181.17–297.75]); 7–12 months, 115.72 (95% CI [89.69–141.75]); 13–24 months, 104.70 (95% CI [61.96–147.44]); 25–36 months, 75.18 (95% CI [43.94–106.42]); and 37–48 months, 33.89 (95% CI [24.43–43.35]). The FC levels demonstrated a clear age-dependent decline, peaking in early infancy (6–12 months) before progressively decreasing with age. Forest plots depicting the pooled values for each age subgroup are shown in Fig. 3. To further explore the age-related trend in FC levels, we conducted a meta-regression analysis using a least squares fractional polynomial model without smoothing. Recognizing potential limitations in analyzing discrete age intervals during early development, we additionally presented the results graphically to illustrate the overall trend (Fig. 2B). The observed pattern indicates a steady reduction in FC levels over the first 4 years of life, underscoring the dynamic nature of intestinal inflammation markers during early childhood.

## Subgroup analysis

To investigate potential sources of heterogeneity in pooled FC reference values among neonates, subgroup analyses were conducted based on feeding type, delivery mode, assay methodology, and geographical region. Detailed results are provided in Table 2 and Fig. S4.

## Feeding type

In terms of feeding type, neonates were classified as breastfed (nine studies, $n = 519$; FC: 140.22 μg/g, 95% CI [96.82–183.62]), formula-fed (nine studies, $n = 372$; FC: 158.73 μg/g, 95% CI [109.96–207.49]), or mixed-fed (two studies, $n = 200$; FC: 150.87 μg/g, 95% CI

**Table 2 Subgroup analyses of the pooled FC reference ranges in neonates.**

| Subgroup | Number of studies (n) | Number of participants (n) | Pooled FC (μg/g) reference ranges (mean, 95% CI) | $Q_{within}$ | $Q_{between}$ | $p$ |
|---|---|---|---|---|---|---|
| Type of feeding | | | | | | |
| Breast-fed | 9 | 519 | 140.22 [96.82–183.62] | 8.79 | 0.48 | 0.788 |
| Formula-fed | 9 | 372 | 158.73 [109.96–207.49] | 6.47 | | |
| Mixed-fed | 2 | 200 | 150.87 [120.69–181.05] | 1.29 | | |
| Delivery mode | | | | | | |
| NSVD | 4 | 239 | 144.86 [126.65–163.08] | 0.99 | 0.58 | 0.467 |
| CSD | 4 | 437 | 155.05 [52.00–403.00] | 7.93 | | |
| Assay methodology | | | | | | |
| LFIA | 1 | 43 | 308.00 [261.50–354.50] | 0 | 6.63 | 0.011 |
| ELISA | 11 | 683 | 298.58 [238.13–359.03] | 36.59 | | |
| FEIA | 3 | 341 | 209.76 [53.64–365.88] | 3.54 | | |
| Region | | | | | | |
| Europe | 9 | 493 | 265.63 [191.92–339.34] | 53.08 | 1.11 | 0.388 |
| Africa | 1 | 21 | 345.00 [132.00–558.00] | 0 | | |
| Asia | 4 | 472 | 241.45 [88.00–546.00] | 11.84 | | |
| North America | 1 | 38 | 122.00 [87.50–156.50] | 0 | | |

Note:
Subgroups analyses are based on a fixed effects model. FC, Fecal calprotectin; CI, confidence interval; NSVD, normal spontaneous vaginal delivery; CSD, cesarean section delivery; LFIA, lateral flow immunoassay; FEIA, fluorescence enzyme immunoassay; $Q_{within}$, homogeneity statistics within subgroups; $Q_{between}$, homogeneity statistics across subgroups.

[120.69–181.05]). No statistically significant differences were observed across groups ($Q_{between}$ = 0.48, $p$ = 0.788), suggesting that feeding type does not have a marked influence on FC levels in neonates.

### Delivery mode

With respect to delivery mode, data from four studies on normal spontaneous vaginal delivery ($n$ = 239) and four studies on cesarean section delivery ($n$ = 437) yielded mean FC levels of 144.86 μg/g (95% CI [126.65–163.08]) and 155.05 μg/g (95% CI [52.00–403.00]), respectively. The difference was not statistically significant ($Q_{between}$ = 0.58, $p$ = 0.467), indicating that the mode of delivery exerts limited influence on neonatal FC levels.

### Assay methodology

In contrast, considerable heterogeneity was detected across assay methodologies. FC levels varied notably depending on the analytical method used: lateral flow immunoassay (one study, $n$ = 13) reported 308.00 μg/g (95% CI [261.50–354.50]); enzyme-linked immunosorbent assay (ELISA; 11 studies, $n$ = 683) reported 298.58 μg/g (95% CI [238.13–359.03]); and fluorescence enzyme immunoassay (FEIA; three studies, $n$ = 341) reported 209.76 μg/g (95% CI [53.64–365.88]). The difference among assay types was statistically significant ($Q_{between}$ = 6.63, $p$ = 0.011), confirming that assay methodology substantially contributes to inter-study variability in FC levels. Within the ELISA subgroup, pronounced heterogeneity ($Q_{within}$ = 36.59) led to further stratification by the
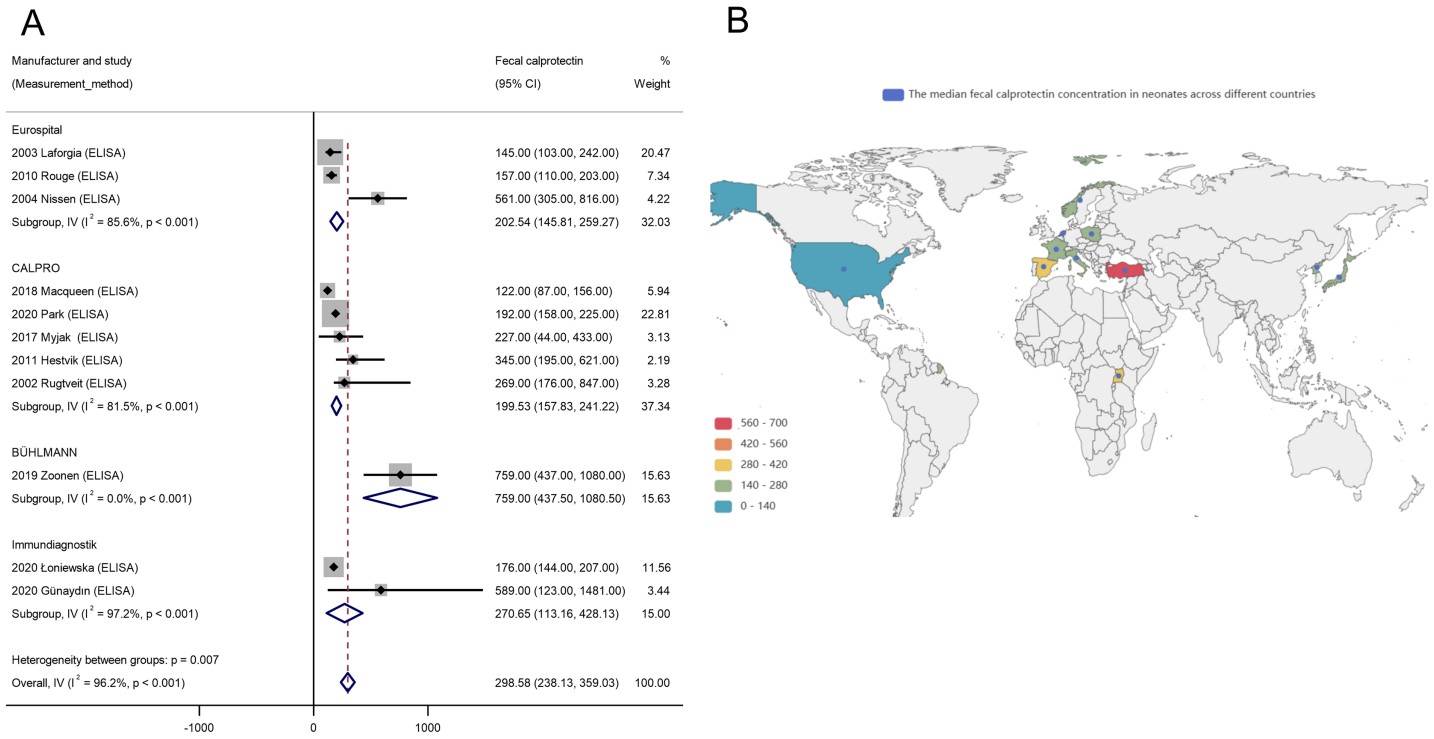

**Figure 4 Influence of ELISA manufacturer and geographical location on neonatal fecal calprotectin levels.** (A) Forest plot depicting subgroup analysis of FC levels stratified by ELISA manufacturer; (B) Global distribution map illustrating the median FC concentration ranges in neonates across different countries.

reagent manufacturer. As depicted in Fig. 4A, different ELISA manufacturers yielded highly variable FC levels, with an overall $I^2$ of 96.2% ($p < 0.01$), highlighting significant methodological variability even within the same assay type.

## Geographical region

Geographical variation was also assessed. Pooled FC levels by region were: Europe (nine studies, $n = 493$; FC: 265.63 µg/g, 95% CI [191.92–339.34]), Africa (one study, $n = 21$; FC: 345.00 µg/g, 95% CI [132.00–558.00]), Asia (four studies, $n = 472$; FC: 241.45 µg/g, 95% CI [88.00–546.00]), and North America (one study, $n = 38$; FC: 122.00 µg/g, 95% CI [87.50–156.50]). Although no statistically significant regional differences were found ($Q_{between} = 1.11$, $p = 0.388$), the global distribution map (Fig. 4B) visually underscores substantial variability in FC levels across countries. For example, Turkey exhibited notably elevated FC levels (marked in red), whereas some parts of Europe and the United States displayed lower values (marked in green), indicating that regional factors may influence FC levels despite the absence of statistically significant heterogeneity.

## DISCUSSION

In this meta-analysis, we established age-specific reference ranges for FC levels in healthy children aged 0–4 years. Our approach offers two significant strengths: first, increased precision through the aggregation of a large pooled sample comprising thousands of

participants; and second, improved internal and external validity resulting from diverse geographic representation and the inclusion of various FC measurement techniques.

The reference intervals derived from 2,556 healthy children provide a robust foundation for defining reliable, age-specific FC benchmarks. Our analysis revealed a clear and gradual decline in mean FC levels (expressed in µg/g) throughout early childhood. FC levels decreased from 289.85 (95% CI [202.79–376.91]) in neonates under 1 month of age to 33.89 (95% CI [24.43–43.35]) by 37–48 months. Notably, a transient elevation was observed between 6 and 12 months of age, with mean levels peaking at 250.27 (95% CI [192.88–307.67]) before steadily declining and approaching adult-like levels by age 4. This age-related trend likely reflects multiple overlapping physiological and immunological maturation processes (*Fagerberg et al., 2003*). While neonates are born with functional gastrointestinal systems, substantial postnatal development continues throughout the first few years of life. During this period, intestinal lengthening progresses, and the gut microbiota transitions toward a stable, adult-like configuration (*Park et al., 2020*). Simultaneously, the secretory immunoglobulin A system—a crucial component of the mucosal immune defense that binds to and neutralizes antigenic stimuli—undergoes gradual development, typically achieving full functional maturity around 4 years of age (*Rougé et al., 2010*). These developmental milestones coincide with the observed stabilization of FC levels and the maturation of the intestinal mucosal barrier (*Roca et al., 2020*).

The consistent downward trajectory of FC levels from infancy through early childhood underscores the dynamic nature of gastrointestinal and immune development during this formative period. These findings carry critical clinical relevance, reinforcing the need for age-specific FC reference ranges in pediatric diagnostic assessments. The temporary peak in FC levels observed between 6 and 12 months may reflect increased mucosal immune activation or transient elevations in intestinal permeability. This phase likely coincides with key physiological transitions, such as the introduction of complementary foods and heightened exposure to environmental antigens. The subsequent decline in FC levels suggests progressive gut maturation, stabilization of the intestinal microbiota, enhanced mucosal barrier integrity, and the gradual refinement of immunoregulatory mechanisms— all of which contribute to reduced baseline intestinal inflammation as children grow older.

These findings highlight the critical need to interpret FC levels within the context of age-related physiological maturation, particularly when using FC as a biomarker to assess gastrointestinal health in pediatric populations. The lower bounds of the CIs for each age group may serve as indicators of intestinal mucosal maturation during early development, while the upper bounds can function as thresholds for distinguishing inflammatory processes. Elevated FC levels may reflect conditions such as autoimmune enteropathy or inflammatory colitis, which are typically managed with anti-inflammatory agents (*Cumpston et al., 2019*; *Sideri, Papageorgiou & Eliades, 2018*). In contrast, persistently low FC levels may be associated with structural abnormalities such as microvillus atrophy or epithelial dysplasia—conditions for which effective therapies remain limited and may require prolonged parenteral nutrition or even intestinal transplantation (*Kolho & Alfthan, 2020*; *Velasco Rodríguez-Belvís et al., 2020*; *Campeotto et al., 2004*). Given the critical role of

FC in informing clinical decision-making—including the determination of the necessity for further diagnostic investigations or referral to specialist care (*Zeng et al., 2025*, *2023*, *2024*)—the establishment of robust, age-specific reference intervals becomes highly valuable.

Subgroup analyses revealed no statistically significant differences in FC levels based on delivery mode or feeding type. While some studies (*Lee et al., 2017*; *Li et al., 2014*; *Savino et al., 2010*) suggest that vaginally delivered and breastfed infants tend to exhibit higher FC levels than those born *via* cesarean section or formula-fed, our findings did not replicate this association. It has been proposed that artificially fed infants may lack essential bioactive components present in human milk that contribute to intestinal tight junction closure, or they may experience mucosal irritation from formula feeding, thereby increasing intestinal permeability. Breastfeeding fosters the growth of beneficial microbial taxa, such as *Bifidobacteria* and *Lactobacilli*, which promote gut maturation, enhance IgA production, regulate T helper cell responses, and contribute to the establishment of a balanced intestinal microbiome. At birth, the fetal gastrointestinal tract is sterile, but it rapidly undergoes microbial colonization influenced by the mode of delivery. Vaginal birth exposes the neonate to maternal vaginal and intestinal microbiota, promoting early gut colonization that supports immune development, metabolic programming, and homeostasis of the host–microbe relationship. Indeed, delivery mode is a key determinant of early microbiome composition, with cesarean section associated with delayed and altered colonization patterns. Although our analysis did not yield statistically significant differences across subgroups, this outcome is broadly consistent with findings from previous investigations (*Velasco Rodríguez-Belvís et al., 2020*; *Campeotto et al., 2004*; *Günaydın et al., 2020*). The apparent discrepancies may be explained by variations in sample size, geographic location, dietary practices, and hygiene standards, underscoring the need for larger, well-controlled studies to further explore these associations.

Nonetheless, there are several limitations to this meta-analysis, which must be acknowledged. Most notably, standardized 95th percentile variance data for FC levels were not consistently available across the included studies. As a result, we employed an alternative approach to estimate reference intervals using the reported median or mean FC levels in conjunction with either standard deviation or range data. This method necessitated a more complex adaptation of the original formulae typically used for calculating effect size variance (Fig. S2).

Another important limitation of this meta-analysis arises from the inherent constraints of the available data. Although we defined discrete age categories to facilitate the synthesis of appropriate reference intervals, the FC data reported in the included studies lacked sufficient granularity to enable more refined age stratification. As a result, we were unable to generate robust reference ranges for narrower age brackets due to data limitations. Furthermore, while the application of strict inclusion criteria enhanced the methodological rigor and internal validity of our analysis, it also necessitated the exclusion of potentially relevant studies that did not fully meet these predefined standards.

Subgroup analyses were conducted to examine potential sources of heterogeneity in neonatal FC reference ranges across variables such as feeding type, delivery mode, assay

methodology, and geographical region (*Roca et al., 2020*; *Herrera, Christensen & Helms, 2016*; *Kapel et al., 2010*). Despite these stratifications, significant heterogeneity persisted, particularly in relation to assay methodology. The uneven distribution of studies across assay types—such as the presence of only one study employing LFIA—and the variation among ELISA manufacturers introduced substantial variability, thereby limiting the reliability of the pooled reference ranges. Geographical representation also emerged as a noteworthy limitation. Compared with Europe (nine studies) and Asia (four studies), data from Africa and North America were severely underrepresented, with only one study from each region. This geographic imbalance may bias regional comparisons and reduce the generalizability of our findings to underrepresented populations, a concern further illustrated by the global distribution map (Fig. 4B), which highlights substantial disparities in participant representation. Moreover, our analysis did not adjust for several potentially influential confounding variables, such as gestational age, birth weight, and maternal health status—all of which could contribute to inter-individual variability in FC levels. Consequently, the findings from our subgroup analyses should be interpreted with caution. These limitations underscore the need for further large-scale investigations using standardized methodologies and more granular data, including comprehensive geographical stratification, to improve the accuracy and applicability of neonatal FC reference intervals.

## CONCLUSIONS

To the best of our knowledge, this is the first systematic meta-analysis to establish age-specific reference ranges for FC in healthy children aged 0–4 years using data pooled from multiple studies. By offering a clearer understanding of FC levels in this population, our findings provide a valuable tool for clinicians to differentiate between normal physiological changes and pathological inflammation, thereby supporting more precise and effective diagnostic and therapeutic strategies in early childhood. However, the subgroup analyses did not account for critical confounding factors such as gestational age, birth weight, and maternal health, which may have influenced FC levels. These limitations highlight the urgent need for future research that adopts harmonized protocols and includes broader geographical representation and finer demographic stratification to establish more accurate and generalizable FC reference intervals in pediatric populations.

### Funding

This study were supported by the National Natural Science Foundation of China Grants No. 8187060357 (to Xiujun Pan) and Shanghai Municipal Health Commission Clinical Research Project No. 202340054 (to Yunlan Zhou) to cover the article processing charges. The funders had no role in study design, data collection and analysis, decision to publish, or preparation of the manuscript.

## Grant Disclosures

The following grant information was disclosed by the authors:
National Natural Science Foundation of China: 8187060357.
Shanghai Municipal Health Commission Clinical Research: 202340054.

## Competing Interests

The authors declare that they have no competing interests.

## Author Contributions

- Junxiang Zeng conceived and designed the experiments, performed the experiments, analyzed the data, prepared figures and/or tables, and approved the final draft.
- Wenxian Yu conceived and designed the experiments, performed the experiments, analyzed the data, prepared figures and/or tables, and approved the final draft.
- Xiupan Gao performed the experiments, analyzed the data, prepared figures and/or tables, and approved the final draft.
- Youyou Yu performed the experiments, analyzed the data, prepared figures and/or tables, and approved the final draft.
- Yunlan Zhou conceived and designed the experiments, prepared figures and/or tables, authored or reviewed drafts of the article, and approved the final draft.
- Xiujun Pan conceived and designed the experiments, authored or reviewed drafts of the article, and approved the final draft.

## Data Availability

  The raw data is available in the Supplemental File.

## Supplemental Information

Supplemental information for this article can be found online at http://dx.doi.org/10.7717/peerj.19572#supplemental-information.

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
