# Peer review of "Establishing reference values for age-related fecal calprotectin in healthy children aged 0–4 years: a systematic review and meta-analysis"

_PeerJ, doi:10.7717/peerj.19572_

## Round 0.1 · original submission · Major Revisions

Please refer to the attached reviews.

**Language Note:** The review process has identified that the English language must be improved. PeerJ can provide language editing services - please contact us at [email protected] for pricing (be sure to provide your manuscript number and title). Alternatively, you should make your own arrangements to improve the language quality and provide details in your response letter. – PeerJ Staff

·

Basic reporting

The challenge (calprotectin being highly variable in young children) is well elucidated and described. The need for the study is clear and apparent, and presented with straightforward language.

Experimental design

Primary concern: not all calprotectin assays are completed the same way, with very clear shifts in reported range (in particular if urea is used in the process, which will elevate results). as y'all have identified the methods used in this approach, you should be able to fairly easily complete an analysis assessing whether method shifted results.

secondary concern: days of diarrheal illness vary by region, which may have an impact on baseline calprotectin due to recent gastrointestinal infection. This would be best addressed by assessing the impact of location on level to ensure that there's no geographic effects.

search criteria are solid, and otherwise design is appropriate, but as it is these results are too confounded.

Validity of the findings

As presented, these results are too confounded. they need to ensure no corrections are needed for methodology of Calprotectin assay and region.

This a lovely and useful idea, that I hope is completed! by making these adjustments

Reviewer 2 ·

Basic reporting

In the submitted manuscript “Establishing reference values for age-related fecal calprotectin in healthy children aged 0-4 years: a systematic review and meta-analysis“ by Zeng et al, the authors aim to establish references ranges for fecal calprotectin for different ages in young children. The manuscript is overall interesting. However, there are major concerns, one being that this meta-analysis does not appear to be comprehensive. E.g. the following articles included in other reviews (e.g. PMID: 20818270) were not included in this meta-analysis: Baldassarre et al x2, Olafsdottir et al, Laforgia et al, Dorosko et al, Rhoads et al. There are certainly various other original articles on that topic that could be identified and included in the meta-analysis. Also, the way the results section is written in the abstract is confusing. It would be better to write mean for age <1 month 289.85 (95% CI 202.79 to 376.91) etc

- L. 178-179: “For more detailed results, please refer to the Supplemental Fig.S4.” This is not appropriate. All results, including those from supplementary figures should be discussed in the results section of the manuscript. In particular, these are the main results of the study. Why are they not a main figure and discussed in detail? Would remove table 2 and figure 2 instead. Those results should also be included in the abstract, as discussed above.
- Would also include suppl figure S5 as a main figure and remove table 3.
- L. 177: mean FC 196.7 vs 77.1 is not a small difference, it is large. Either it should say “large”, or better provide a percent or fold difference between the groups, as this is more objective. Ideally, the result section should only contain objective data, whereas those results can be discussed/judged in the discussion portion.
- In the abstract, it is written that databases were searched through December 2020 but in the methods, the final update was done in April 2024. Please adjust.
- It would be interesting to see if there are indications that different countries or continents have (markedly) different calprotectin ranges. Please include as a sub-analysis.
- Also, please compare the values obtained by different methodologies, as this likely has an impact on the values.
- L. 45: Would avoid the term ‘trimester’ here, as this usually refers to pregnancy.
- L. 46: “could exert beneficial [effects on the] host defense”
- In suppl fig S2, the arrows going away from “Median available?” on the left would have to be changed as both say “No” but show 2 different results.
- L. 190-192: The results show that the mean values steadily decrease from <1 month through 3-4 years. This should also be stated and discussed as such in the discussion.

Experimental design

see above

Validity of the findings

see above

Additional comments

-

Reviewer 3 ·

Basic reporting

In general, the English language in this manuscript needs to be improved. Before it can be published, I would suggest further proofreading to enhance clarity and ensure grammatical accuracy. Please check the following examples of grammatical errors/typos:
• Line 78 what is “initial hurdle”?
• Line 116-117
• Line 121-122
• Line 124 “express the finding” should be “present the findings”, and what findings? Please rephrase
• Line 128 please rephrase “was regarded as significant differences, and”
• Line 145 “age groups” should be “age group”
• Line 145
• Line 205-206 grammatical error. May be rephrased to “including the determination of necessity for ....”
• Line 221
• Line 225: “inclusion literature” should be “literature included in the meta-analysis”
• Line 235-236 what do you mean by “we grouped according to specific months of the merger,”?

Some other comments about the reporting:
• In terms of current research of the area, line 50-52 the authors should comment more on the current literature/research on the age-related reference values for FC, showing why these studies have inconsistencies/contradictions
• Line 30-34: may consider using bullet point to present the results
• There are several incorrect Figure/table numbers, for example Line 109, line 165, 180, 229.
• Line 240-241 please clarify why the impact is small
• Line 119 add reference for tool: “JBI-MAStARI”
• Line 154-155 clarify “level 1 and level 2” in the main text, although they are explained in the table footnote
• Line 225: please clarify if it is percentile value or percentile’s variance, which is not consistent with line 105 (percentile's variance)

Experimental design

Statistical analysis:
• Line 104 please clarify if frequentist random effect model by reference # 13 was used when you say “normal approximation method”
• Line 106-107 clarify what alternative approach is
• Line 126 to 127 please clarify the random-effects model used for meta analysis and add a reference. is it the one mentioned above in reference 13?
• Line 137-138 why is there only one study through citation search? Clarify why this was not in the 74 studies?
• Line 141 IQR is a range, but it looks that 25th and 75th percentiles are presented (IQ, interquartile), for example, in the first row of the table, please clarify

Validity of the findings

• According to reference 13, one of the key assumptions of the frequentist method is that the true within-study variances are the same in all studies and that any observed differences are due to sampling variability. Did the authors evaluate this assumption?

---

## Round 0.2 · accepted · Accept

All the comments were addressed.

·

Basic reporting

well reported, easily legible, and professionally done.

Experimental design

SUBSTANTIAL improvement with the subgroup analyses, and the new significant findings go a tremendous way to making this paper more meaningful AND novel.

Validity of the findings

well executed.

Additional comments

Authors did a remarkable job incorporating feedback.

Reviewer 2 ·

Basic reporting

No comment

Experimental design

No comment

Validity of the findings

No comment

Additional comments

No comment

Reviewer 3 ·

Basic reporting

No additional comments as all comments from previous review have been addressed.

Experimental design

No additional comments as all comments from previous review have been addressed.

Validity of the findings

No additional comments as all comments from previous review have been addressed.